# Help-seeking in older crime victims: A mixed-methods study in collaboration with the Metropolitan Police Service

**Marc Serfaty**[1,2]*, **Jo Billings**[1], **Victoria Vickerstaff**[3], **Teresa Lee**[3], **Marta Buszewicz**[4], **Jessica Satchell**[1]

**1** Division of Psychiatry, University College London, London, United Kingdom, **2** Priory Hospital North London, London, United Kingdom, **3** Research Department of Primary Care and Population Health, University College London, London, United Kingdom, **4** Great Ormond Street Institute of Child Health, University College London, London, United Kingdom

\* m.serfaty@ucl.ac.uk

## Abstract

### Background

There are growing concerns about the psychological impact of community crimes on older victims, but little is known about whether older victims obtain mental health support.

### Objective

To understand: A) whether older crime victims seek help for psychological distress, B) what factors predict help-seeking, and C) the barriers and facilitators to accessing support.

### Methods

Our longitudinal mixed-methods study was embedded within the Victim Improvement Package (VIP) trial. Older victims (n = 2,932) were screened for depressive and anxious symptoms with the GAD-2 and PHQ-2 within one month of a crime. Those with significant symptoms (n = 1,170) were provided with letters signposting them to their General Practitioner (GP) (Family Physician). A subsample of older Victims (n = 677) were then re-screened at three months and asked if they had acted on the signposting. Logistic regression was used to examine predictors of help-seeking. Qualitative semi-structured interviews on a sub-sample (n = 27) were undertaken to establish barriers and facilitators to help-seeking and explore views on the signposting letter, and analysed using thematic analysis.

### Results

Only 13% (n = 85) of distressed older victims approached their GP for help, and only 32% (n = 27) of these received help. Significant predictors of acting on signposting were police-recorded vulnerability (p = 0.01) and severity of continued anxiety at three months' post-crime (p <0.01). Help seeking appeared to be driven by feeling overwhelmed or a desire to find others with similar experiences. Barriers to help-seeking included accessibility problems and the belief that they should be able to cope.

**Data Availability Statement:** Supporting data for the quantitative component is available from the corresponding author or priment@ucl.ac.uk on

request. Supporting data for the qualitative component is not available as the information contained may compromise the privacy of research participants, who did not agree to their data being shared publicly.

**Funding:** The research was funded by the NIHR-PHR Victim Improvement Package Trial (grant number 13/164/32). JS and MS are also supported by the NIHR University College London Hospitals Biomedical Research Centre at University College London Hospitals NHS Foundation Trust and University College London. The views expressed are those of the authors and not necessarily of the NIHR or the Department of Health and Social Care. The funders had no other role other than financial support.

**Competing interests:** The authors have declared that no competing interests exist.

## Conclusions

Despite growing evidence of psychological distress in older crime victims, few receive support. Signposting older victims may be insufficient to improve psychological outcomes and help-seeking barriers suggest more active management is required.

## 1. Background

There are growing calls to recognise the psychological impact of crime in older victims as a public health concern [1, 2]. Recent data for England and Wales reports that almost a fifth (19%) of the population is aged 65 and over [3] and around 10% of crimes committed between 2022 to 2023 affected this age group [4]. Experiencing a crime in older adulthood appears to be associated with poor mental health outcomes [1, 5]. Depression, anxiety, and post-traumatic stress disorder have been found in older victims of all crime types [2]. In a UK study of 486 older victims of community crime, 27% of victims suffered distress symptoms for at least three months, with roughly half meeting criteria for psychiatric disorder [2]. Studies to assess the effectiveness of psychological interventions for older crime victims are underway [6], however, little is known about engagement with mental health services in this population. Utilisation of mental health services is often low in older adults, who often interpret psychiatric symptoms as a normal response to ageing and life stressors [7]. However, limited data exists on whether older adults view their symptoms and mental health services differently if their distress is attributed to a crime. A lack of awareness of primary care services and feelings of shame and embarrassment have been linked to low formal help-seeking rates for victims of all ages [8], leading to informal help-seeking via friends, family, and their community [9].

Awareness and accessibility of formal help-seeking services may be facilitated by 'signposting' victims to services [8, 10]. Police, as the initial responders, may be best placed to detect needs and assist in signposting victims to appropriate services in a community setting (primary care). According to Goldberg's Filter Model [11], the steps to improving the health of a population are 1) to increase awareness of impact, 2) improve case detection, 3) increase referrals of those detected, and 4) improve treatment of those referred. Victims vary in the degree of psychological distress they experience after a crime, so it is important to target support to those most in need [12].

Resources are currently limited for determining the levels of distress and the needs of older crime victims. Detecting significant distress early, followed by action from formal help-seeking services may improve psychological outcomes for older crime victims [12]. Contact with victim service agencies has been shown to be a powerful predictor of formal help-seeking behaviour in adult victims of violent crime the USA [12]. Whether signposting improves service utilisation in older victims of different crime types the United Kingdom remains unclear. Little is also currently known about the support offered to older victims who do seek help and whether they perceive this as useful and acceptable. As the recent Health and Care Act [13] recommends joining up healthcare with public services, we collaborated with the Metropolitan Police Service to understand help-seeking in this group and assess the effectiveness of early screening and signposting to primary care services.

### Study aims

Our study used mixed methods to address the following:

**Quantitative.**

1. To generate descriptive statistics on older victims' help-seeking behaviour.

2. To determine the predictors of help seeking in older crime victims' of crime.

**Qualitative.**   To qualitatively explore what older victims consider the barriers and facilitators to help-seeking and their views on the signposting letter in a subset of older victims selected from the quantitative study.

## 2. Methods

### Study design and setting

We conducted a mixed-methods longitudinal and interview study, embedded within a clinical trial, assessing the clinical and cost-effectiveness of cognitive behaviour therapy for older crime victims. We collected data between 9th November 2017 and 12th September 2019 across nine London boroughs selected for the highest crime rates, proportion of older adults, and range of sociodemographic characteristics.

### The Victim Improvement Package trial

Our study was nested within the Victim Improvement Package (VIP) Trial (ISRCTN: 16929670), full details for which are published elsewhere [6]. In this three-step trial, participants were identified as distressed (step 1) within two months of a crime, rescreened for distress (step 2) at 3 months of the crime and then if suitable randomised into a single-blind randomised controlled trial (step 3) comparing treatment as usual against treatment as usual plus up to 10 sessions of a manualised Victim Improvement Package (VIP). Contact with crime victims was provided through Safer Neighbourhood Teams. Safer Neighbourhood Teams are small teams of police and community support offers who serve a defined geographical area. Safer Neighbourhood Teams screened older victims using screening measures for anxiety (GAD2) and depression (PHQ2) within two months of a community crime. Those who scored positive for depression and/or anxiety were given signposting letters to give to their General Practitioner (GP); a GP or Family Physician is a physician who treats acute and chronic illnesses and provides preventive care and health education to patients of all ages, and who works primarily in a community setting. Older victims were followed up at step 2 to establish their use of the letters and identify eligible participants for qualitative interviews.

### Inclusion criteria

Inclusion criteria: English-speaking older adults aged 65 years or more who had reported property, violent, harassment, theft, or fraudulent crimes to the Metropolitan Police Service. Sexual crimes and domestic violence were excluded because the additional complexities associated with these crimes warrants specialist management.

### Quantitative methods

**Signposting older victims to their General Practitioner (GP).**   Participants were recruited by Safer Neighbourhood Teams dealing with community-based offences during home visits within 2 months of the crime. In addition to their usual advice and support, Safer Neighbourhood Teams screened older victims for depressive and anxious symptoms using the GAD-2 and PHQ-2 [14, 15] after obtaining consent to share the university screening data with the research trial team. All victims were provided with information about the potential impact

of crime. Screen-positive victims were given a letter signposting them to their GP, signed by the VIP trial chief investigator. The letter stated they had suffered a crime and screened as significantly distressed on the GAD-2 and PHQ-2 and suggested the GP should manage them as they deem appropriate. Participants who screened negative were also given a GP letter making them aware of potential symptoms that could present following a crime and suggested that if they experienced further distress, they should approach their GP. University researchers followed-up with consenting screen positive older victims within three months of crime through home visits or telephone and were re-screened for continued symptoms on the GAD-2 and PHQ-2, and also asked whether they had given the letter to their GP and what help they had received, if any.

**Measures.** Depressive and anxious symptoms at one month (screening) and three months (re-screening) post-crime were assessed using a four question pro-forma consisting of the PHQ-2 [14] and GAD-2 [15], with participants asked to rate how much they felt bothered by depressive and anxious symptoms over the previous two weeks on a 4-point scale. Caseness was determined by a score of $\geq 3$ on the PHQ-2 and $\geq 2$ on the GAD-2, as recommended for use in older adults [16]. The PHQ-2 and GAD-2 have been used previously in older victims [2, 9] and their brevity and simplicity meant they could be reliably administered by Safer Neighbourhood Teams with a half day's training.

**Descriptive statistics.** Using a one-page questionnaire, data were collected on older victims' sociodemographic characteristics, living arrangements, social networks, and crime characteristics. At re-screening, the questionnaire was extended to ask initially screen-positive older victims whether they remembered receiving the signposting letter (yes/ no), whether they gave the letter to the GP (yes/ no) and, if yes, what action the GP took (free response). For categorical data, the number of observations and percentages will be presented. For continuous data, means and SDs will be presented under the assumption that the data is normally distributed.

Help-seeking behaviour was measured by whether screen positive participants acted on their signposting letters. Among participants who remembered receiving the signposting letter, demographic and crime related characteristics were summarised using mean, standard deviation, count and percentage where applicable.

**Analysis of predictors of acting on signposting in older victims.** Demographic data collected by Safer Neighbourhood Teams at initial screening were analysed using univariable logistic regression to determine predictors of acting on signposting. Given the dearth of research determining help-seeking in older victims of community crime, we used a predominantly data-driven approach, although we did consider variables known to be associated with access to health care and help-seeking in other victim groups [8, 12]. These included: age, gender, ethnicity, marital status, education, living arrangements, number of social contacts, previous depression, Crime Reporting Information System score (a measure of vulnerability), crime type, whether they were a previous victim, whether perpetrator arrested, daily life affected, sense of safety before and after crime, severity of depressive symptoms at step 1, severity of anxious symptoms at step 1, and screening outcome at Step 2.

Participants' responses to whether they had given the letter to their GP were categorised into a binary outcome variable (yes/no). A small number reported seeing their GP for further help without the letter which we classified as 'yes' as it was still help-seeking. Any responses which could not be classified as yes/no were coded as missing. Logistic regression models were used to investigate the association between participants' characteristics and help-seeking behaviours. Univariable logistic regression models based on P<0.2 were used to screen potential predictor variables on help-seeking behaviours as a method of variable selection. Multivariable logistic regression models and backward elimination method with P<0.05 were used to

select a final set of variables as the predictors. No consensus exists about the best method for selecting the predictor variables, but backward elimination is generally the preferred method [17]. This analysis was conducted using Stata version 17.

## Qualitative methods

**Study of barriers and facilitators to help-seeking.** Twenty-seven semi-structured interviews were conducted with older crime victims to explore the barriers and facilitators to using the GP signposting letter and help-seeking in depth. Purposive sampling was used to achieve a sample that was diverse across gender, age, ethnicity, religion, crime type, positive and negative outcomes on the PHQ-2 and GAD-2 at both screening and re-screening, and use of the GP letter recorded at re-screening. Older victims were invited to participate during re-screening home visits or telephone calls and, if agreeable, a separate interview visit arranged, with the aim of establishing rapport in advance. The interviews were conducted pre-randomisation to the VIP trial.

The interview topic guide, enquired about the older victims' background and details of the crime before asking about their understanding and experience of different support services, reasons why they did or did not seek help, and views about the signposting letter. Questions were open-ended. Interviews were conducted by the researchers through home visits or at the university as preferred by the participant and were mostly one-to-one except for three participants who chose to have a family member present. Interviews ranged from 34–132 minutes and lasted on average 73 minutes, of which an average of 30 minutes was spent discussing help-seeking. Interviews were audio recorded and transcribed verbatim.

## Qualitative analysis

The qualitative interviews were analysed using thematic analysis [18], a theoretically flexible approach widely used in health research [19]. Thematic analysis was conducted by two researchers under the supervision of the core team following the steps outlined by Braun and Clarke [18]. Interviews were separately coded using NVivo 12 [20] and together the team developed the coding framework. In accordance with triangulation methods, the quantitative and qualitative data were collected concurrently, analysed via different methods as described, and the data integrated for interpretation of the results [21]. This allows for an expansion of the quantitative findings via qualitative methods and provides both in-depth and generalisable data.

**Positionality statement.** Our research team is made up of qualified mental health professionals (MS, JS, JB), a GP (MB), and statisticians (VV and TL). Some of our team have been crime victims and reported to the police but have not sought mental health support for this. The authors recognise their previous experiences in mental health research may have influenced their expectations of what might appear in the data.

## Ethics

Our study was approved by the University College London Research Ethics Committee (6960/001) on 17[th] March 2016 and was conducted in compliance with the Declaration of Helsinki. Informed written consent was obtained from all participants.

## 3. Results

### Service utilisation in a quantitative sample of older crime victims

Fig 1 is a flow diagram for the numbers of participants screened (step 1) and rescreened (step 2) for the VIP trial is shown. There were only limited data in the reasons for people not

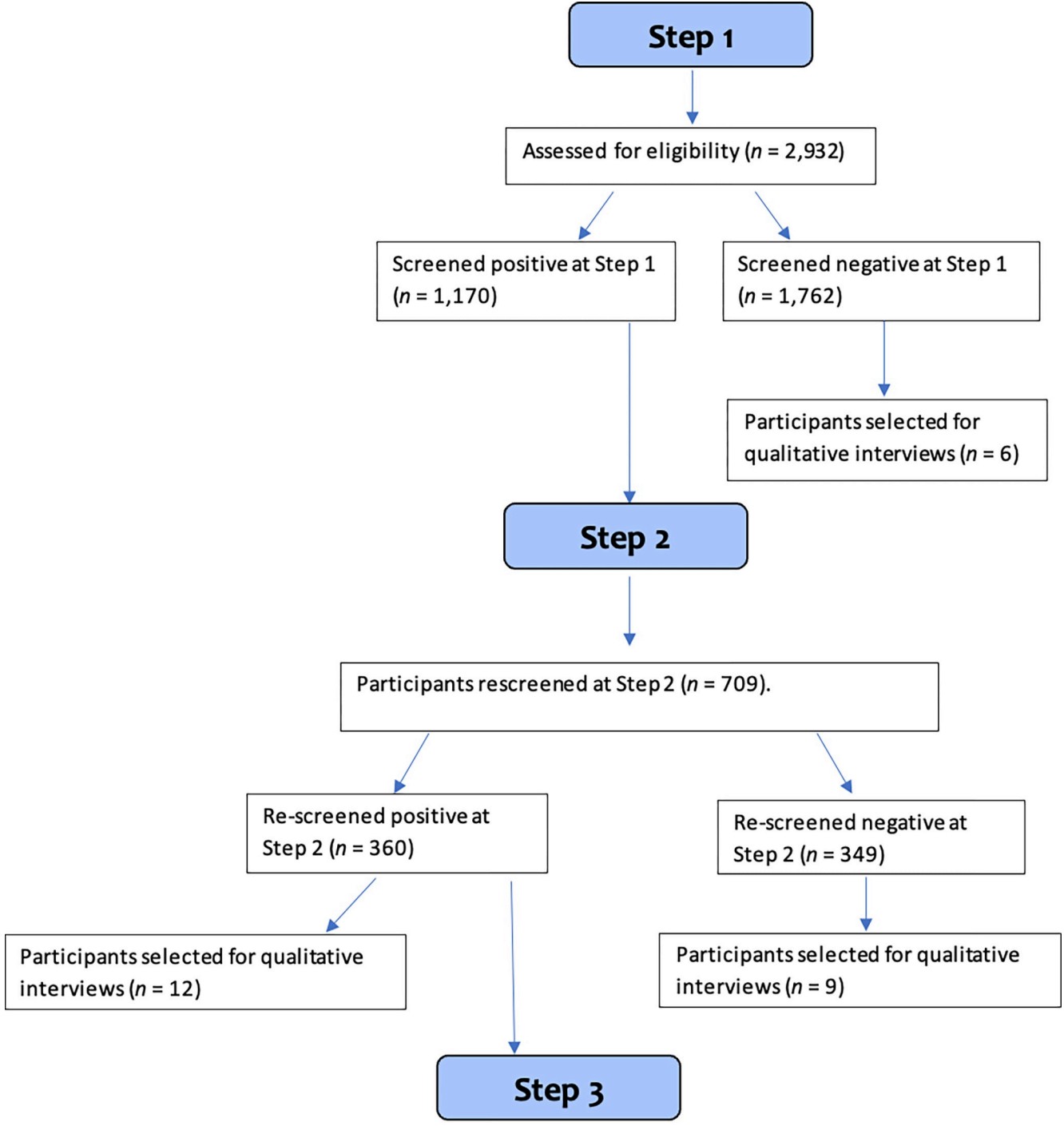

**Fig 1. Flow diagram for VIP trial and qualitative interviews.**

engaging with rescreening (not contactable, incorrect contact details, refused follow-up, declined because of bereavement or poor health). The stage (step 1 or 2) and status (screen positive or negative) of participants for qualitative interviews is also shown in the flow diagram.

Of the 2,932 participants who were initially screened, 1,170 (39.9%) screened positive at step 1. Of these, 709 (60.6%) were followed up at step 2 and of these 677 (95.5%) were asked about signposting, of which 450 remembered receiving information. There were 31 participants (4.5%) not involved as it was during sub-study start up period and one declined. Of those asked about signposting, 342 (51%) re-screened positive on at least one measure.

## Sample characteristics

The sample characteristics of those re-screened and who were asked about signposting (n = 677) are presented in Table 1. At three-month follow-up, 51% (*n* = 342) continued to screen positive for psychological distress on the GAD-2 or PHQ-2.

## Acting on signposting

Fig 2 shows the numbers that acted on signposting. Less than 13% (*n* = 85/677) of screen-positive victims identified at step 1 acted on signposting by visiting their GP (with or without the letter). Even for those who remembered (67%; 450/677) receiving a letter, only 19% (85/450) acted on the signposting letters. Of those who did act, the GP referred for further help less than a third of cases (*n* = 27/85 (32%)). Taken together, less than 4% (27/677) of older victims suffering initial psychological distress after a crime received formal support from healthcare services.

## Predictors of signposting

Logistic regression was used to determine predictors of help seeking. Using univariable logistic regression models, we found that sex, marital status, Crime Reporting Information System scores indicating vulnerability, whether the crime had affective their daily life, previous depression, sense of safety after crime, severity of depressive and severity of anxious symptoms both at step 1 and step 2, and screening outcome at Step 2 were predictor variables with p < 0.2 (Table 2).

Using the backward elimination method with p<0.05, we eliminated sex, marital status, whether the crime had affected their daily life, previous depression, sense of safety after crime, severity of depressive symptoms at step 1 and 2, severity of anxious symptoms at step 2, and screening outcome at Step 2. The final multivariable logistic regression model suggested that only vulnerability (P = 0.01) and severity of anxiety symptoms at step 2 (P <0.01) were associated with acting on signposting (Table 3).

## Qualitative results

**Qualitative sample characteristics.**   Older victims (*N* = 27) from step 1 and 2 participated in qualitative interviews (Fig 1). Participants ranged from 65 to 94 years in age (*M* = 74; SD = 8.05) and 16 (59%) were female. As shown in Table 4, there was a mixture of sociodemographic characteristics, crime types, those who did and did not seek help, and screening outcomes at both step 1 and step 2 in the sample.

**Service use in the qualitative sample.**   Of our 27 qualitative participants, only five had discussed the crime with their GP and two other interviewees took their letter to the GP but did not hear anything further: "*The [doctor] said, 'we'll let you know,' but I haven't got it back. . . it's like nearly two months now.*" (P5). Of those that discussed with their GP, one was referred to counselling, one received practical help in the form of a supporting letter to their housing association, and the other three received no action but said talking to their GP was helpful. Of the remaining participants who did not act on their letter, 14 remembered receiving it, suggesting

**Table 1. Sample characteristics.**

| N = 677 | N | n or Mean | % or SD | N = 677 | N | n or Mean | % or SD |
|---|---|---|---|---|---|---|---|
| **Age** | 667 | 74.4 | 7.3 | **Acted on signposting letter** | 645 | | |
| **Sex** | 674 | | | Yes | | 85 | 13.2 |
| Male | | 250 | 37.1 | No | | 560 | 86.8 |
| Female | | 424 | 62.9 | **Perpetrator arrested** | 668 | | |
| **Ethnicity** | 673 | | | Yes | | 34 | 5.1 |
| White | | 472 | 70.1 | No | | 634 | 94.9 |
| Black | | 65 | 9.7 | **Previous victim** | 675 | | |
| Asian | | 95 | 14.1 | Yes | | 118 | 17.5 |
| Other | | 41 | 6.1 | No | | 557 | 82.5 |
| **Marital status** | 660 | | | **Affected daily life** | 674 | | |
| Divorced/ Separated | | 100 | 15.2 | Yes | | 530 | 78.6 |
| Married | | 262 | 39.7 | No | | 144 | 21.4 |
| Widowed | | 188 | 28.5 | **Weekly social contacts** | 616 | 7.7 | 9.3 |
| Single | | 90 | 13.6 | **Previous depression** | 669 | | |
| Other | | 20 | 3.0 | Yes | | 288 | 43.1 |
| **Education** | 651 | | | No | | 381 | 57.0 |
| Higher degree or equivalent | | 249 | 38.3 | **Sense of safety before crime** | 670 | | |
| Secondary | | 307 | 47.2 | Very safe/ safe | | 521 | 77.8 |
| Primary | | 95 | 14.6 | Neither safe nor unsafe | | 73 | 10.9 |
| **Living arrangement** | 676 | | | Very unsafe/ unsafe | | 76 | 11.3 |
| Rented | | 233 | 34.4 | **Sense of safety after crime** | 670 | | |
| Owner/ occupier | | 419 | 62.0 | Very safe/ safe | | 124 | 18.5 |
| Other | | 24 | 3.6 | Neither safe nor unsafe | | 147 | 21.9 |
| **Crime type** | 677 | | | Very unsafe/ unsafe | | 399 | 59.6 |
| Assault | | 44 | 6.5 | **Severity of anxiety symptoms step 1 (GAD-2)** | 677 | | |
| Burglary | | 169 | 25.0 | < 2 | | 13 | 1.9 |
| Criminal damage | | 49 | 7.2 | ≥ 2 | | 664 | 98.1 |
| Distraction burglary | | 32 | 4.7 | **Severity of anxiety symptoms step 2 (GAD-2)** | 677 | | |
| Fraud | | 36 | 5.3 | < 2 | | 348 | 51.4 |
| Theft | | 213 | 31.5 | ≥ 2 | | 329 | 48.6 |
| Theft with threat | | 122 | 18.0 | **Severity of depression symptoms step 1 (PHQ-2)** | 676 | | |
| Other | | 12 | 1.8 | < 3 | | 363 | 53.7 |
| **Vulnerability** | 676 | | | ≥ 3 | | 313 | 46.3 |
| No recorded vulnerable | | 586 | 86.7 | **Severity of depression symptoms step 2 (PHQ-2)** | 677 | | |
| Recorded vulnerable | | 90 | 13.3 | < 3 | | 477 | 70.5 |
| **Remembering signposting letter** | 677 | | | ≥ 3 | | 200 | 29.5 |
| Yes | | 450 | 66.5 | **Screening outcome at step 2** | 677 | | |
| No | | 227 | 33.5 | Positive | | 342 | 50.5 |
| | | | | Negative | | 335 | 49.5 |

awareness is not enough for help-seeking. When asked for their views on the letter, all of them thought it was a good idea for other people. However, they often did not think it was needed for themselves: "*It's a good idea, you know, to try and help people and that. . . but for me, I suppose I thought it wasn't necessary*" (P22).

There were a few examples of participants who visited their GP because they were distressed: "*I was feeling pain, I went to my doctor. . . you know because every day I cry*" (P23). Some did not seek help simply because they considered it unnecessary: "*If I waltzed in there,*

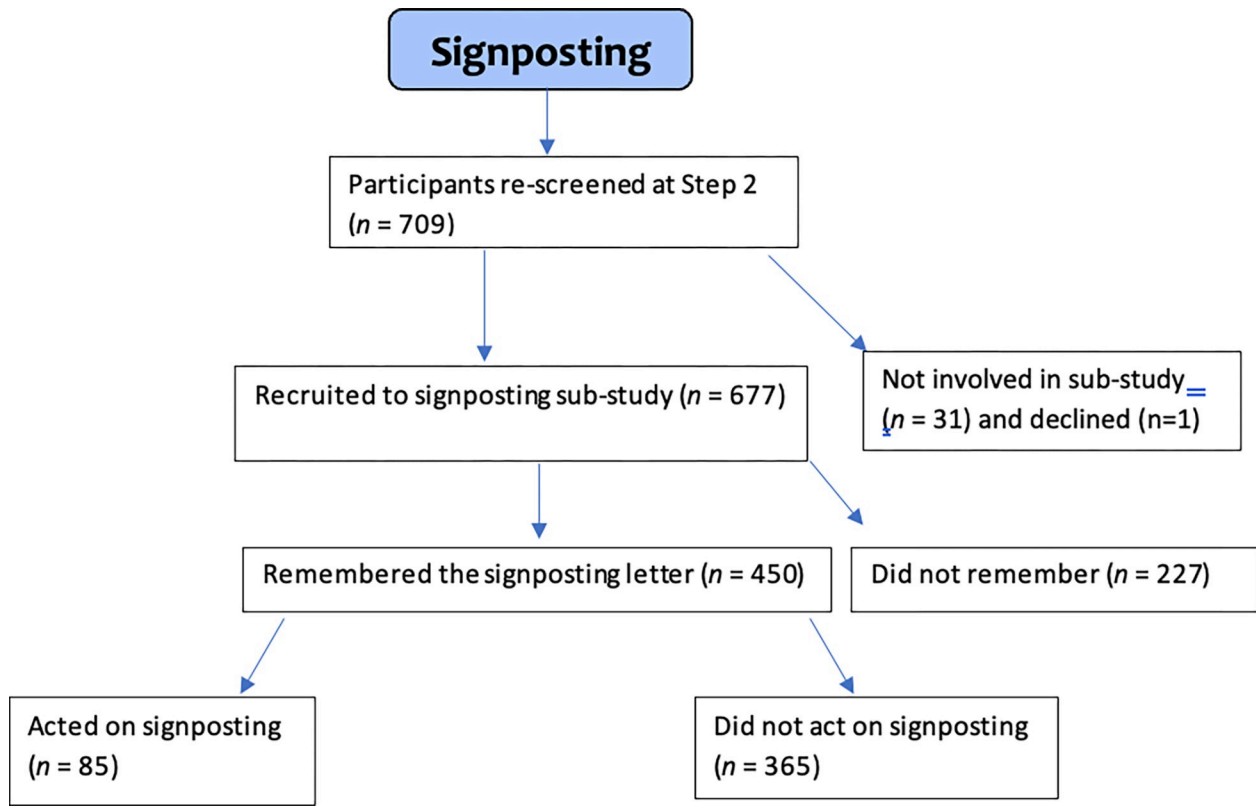

**Fig 2. Flowchart of participants who remembered and acted on signposting.**

*sat down and said, 'I had an attempted burglary, but I feel okay,' she'd go, 'what the hell you doing here?'" (P3).* However, others were distressed but still did not attend: "*I don't fancy running around on antidepressants. . .but I am a bit depressed*" (P21).

## Barriers and facilitators to help-seeking

Through thematic analyses of barriers and facilitators to help-seeking, three themes were developed: older victims' perception of themselves, perception of the crime, and perception of services. Perception of services was further organised into accessibility and acceptability. These themes are summarised in Table 5.

**Perception of themselves.** Some participants did not seek help because they felt personally responsible for the crime. One participant, who had Parkinson's Disease, explained: "*One consequence of the event was feeling I might be increasingly incompetent about things like locking the car. . . So when the boot was empty, £150 quid's worth of stuff, the biggest effect was to say to myself what an idiot I'd been leaving the car door open*" (P9). Feelings of shame and self-blame were also reported by victims of deceptive crimes such as fraud and distraction burglary: "*You feel a bit ashamed that you let it happen without realising."* (P6). Despite feeling distressed, these feelings were barriers to seeking informal support from their friends: "*I felt stupid. . . I didn't tell my friends I had been scammed*" (P15).

Some older victims expressed views that they should '*just get on with it*' (P16). Some felt they should be able to cope by themselves: "*You've got to cope with things, you're on your own and it's up to you*" (P22) even when family were encouraging them to seek help: "*I have emailed her GP to tell her. . . and social services*" (P22—daughter). Some seemed to interpret help-

**Table 2. Predictors of acting on signposting.**

.

| Acted on Signposting Letter (yes/no) | Number of Observations | Odds Ratio | 95% CI | p-value | Acted on Signposting Letter (yes/no) | Number of Observations | Odds ratio | 95% CI | p-value |
|---|---|---|---|---|---|---|---|---|---|
| **Age** | 635 | 1.02 | (0.99, 1.05) | 0.24 | **Vulnerability** | 644 | | | |
| **Sex** | | | | | No recorded vulnerability | | 1.00 | | <0.01 |
| Male | 642 | 1.00 | | 0.05 | Recorded vulnerability | | 2.26 | (1.29, 3.97) | |
| Female | | 1.68 | (1.01, 2.80) | | **Perpetrator arrested** | 637 | | | |
| **Ethnicity** | 642 | | | 0.65 | No | | 1.00 | | 0.56 |
| White | | 1.00 | | | Yes | | 0.69 | (0.21, 2.34) | |
| Black | | 0.72 | (0.30, 1.75) | | **Previous victim** | 643 | | | |
| Asian | | 1.33 | (0.72, 2.47) | | No | | 1.00 | | 0.85 |
| Other ethnic group | | 1.19 | (0.48, 2.96) | | Yes | | 1.06 | (0.58, 1.94) | |
| **Marital status** | 629 | | | 0.06 | **Affected daily life** | 642 | | | |
| Divorced | | 1.00 | | | No | | 1.00 | | 0.01 |
| Married | | 0.56 | (0.27, 1.15) | | Yes | | 2.55 | (1.24, 5.22) | |
| Widow | | 1.18 | (0.59, 2.35) | | **Weekly social contacts** | 587 | 1.00 | (0.97, 1.03) | 0.98 |
| Single | | 1.34 | (0.61, 2.94) | | **Previous depression** | 637 | | | |
| Other | | 0.37 | (0.04, 2.98) | | No | | 1.00 | | 0.02 |
| **Education** | 620 | | | 0.90 | Yes | | 1.77 | (1.12, 2.80) | |
| Higher degree or equivalent | | 1.00 | | | **Sense of safety before crime** | 639 | | | 0.68 |
| Secondary | | 0.95 | (0.47, 1.94) | | Very safe | | 1.00 | | |
| Primary | | 0.89 | (0.54, 1.48) | | Safe | | 0.78 | (0.46, 1.32) | |
| **Living arrangement** | 645 | | | 0.24 | Neither safe nor unsafe | | 0.54 | (0.21, 1.36) | |
| Council rented | | 1.00 | | | Unsafe | | 0.94 | (0.42, 2.14) | |
| Housing assoc rented | | 2.31 | (0.90, 5.89) | | Very unsafe | | 1.21 | (0.25, 5.91) | |
| Private rented | | 1.33 | (0.39, 4.51) | | **Sense of safety after crime** | 639 | | | 0.03 |
| Owner/ occupier | | 1.82 | (0.90, 3.69) | | Very safe | | 1.00 | | |
| Other | | 3.32 | (1.01, 10.91) | | Safe | | 0.64 | (0.18, 2.20) | |
| **Crime type** | 645 | | | 0.26 | Neither safe nor unsafe | | 0.54 | (0.16, 1.82) | |
| Assault | | 1.00 | | | Unsafe | | 0.52 | (0.17, 1.65) | |

(*Continued*)

**Table 2.** (Continued)

.

| Acted on Signposting Letter (yes/no) | Number of Observations | Odds Ratio | 95% CI | p-value | Acted on Signposting Letter (yes/no) | Number of Observations | Odds ratio | 95% CI | p-value |
|---|---|---|---|---|---|---|---|---|---|
| Burglary | | 0.76 | (0.30, 1.92) | | Very unsafe | | 1.29 | (0.40, 4.17) | |
| Criminal damage | | 0.59 | (0.17, 2.04) | | **Severity of depressive symptoms at step 1 (PHQ-2 score)** | 644 | 1.18 | (1.06, 1.32) | <0.01 |
| Distraction burglary | | 1.69 | (0.54, 5.30) | | **Severity of anxiety symptoms at step 1 (GAD-2 score)** | 645 | 1.30 | (1.13, 1.51) | <0.01 |
| Fraud | | 0.63 | (0.17, 2.35) | | **Severity of depressive symptoms at step 2 (PHQ-2 score)** | 645 | 1.16 | (1.04, 1.29) | <0.01 |
| Theft | | 0.48 | (0.19, 1.24) | | **Severity of anxiety symptoms at step 2 (GAD-2 score)** | 645 | 1.15 | (1.04, 1.27) | <0.01 |
| Theft with threat | | 0.94 | (0.36, 2.44) | | **Screening outcome at step 2** | 645 | | | <0.01 |
| Other | | 0.97 | (0.17, 5.44) | | Negative | | 1.00 | | |
| | | | | | Positive | | 1.90 | (1.18, 3.04) | |

seeking as meaning they were less independent: "*I'm what you'd call, really independent. I've had Victim Support before, and it's not helped me, because it's not relevant to me, because I've got my own mind*" (P7) or that it was a sign of weakness: "*The letter is good* [for other people] *because not everybody is strong emotionally and they might fall to pieces not knowing what to do*" (P10).

**Perception of the crime.** There was a common perception that some crimes should be given more professional help than others. Participants would often compare their experiences with other crimes they felt would have been worse. There was no consensus across the sample about which crimes required such help, instead the emphasis was on crimes they had not suffered themselves: "*Having your laptop stolen, you get over it, buy a new laptop. . .. Maybe if they'd set fire or trashed the place*" (P3). This was observed across different crimes, including those that many might consider severe, such as armed robbery: "*I suppose for some a knife might be very traumatic. It wasn't particularly, if they'd waved a gun around, I might have felt very differently about it*" (P19). For some participants, this perspective helped them make peace with their experience. For others, it seemed they were downplaying the impact it had, perhaps because they felt they should not have been affected by it. For example, one older victim (P2) described how ongoing harassment from his neighbour had affected his mood: "*I can't be bothered, I've lost interest. I think it's affected the wife as much as me, we come home, it's a case of 'oh well, we'll go to bed now'*" but when asked whether he had seen his GP, he responded: "*As I said to you, it's not theft, it's not damage, it's just verbal abuse*".

Participants also appeared to think that professional services would share this view "*I didn't tell my GP because I wasn't harmed was I? I was just emotionally. . .you know affected*" (P10).

**Table 3. Multivariable logistic regression results (n = 644) on whether acting on signposting.**

| Outcome = Acted on Signposting Letter (yes/no) | Odds Ratio | 95% CI | p-value |
|---|---|---|---|
| **Vulnerability** | | | |
| No recorded vulnerability | 1.00 | | 0.01 |
| Recorded vulnerability | 2.06 | (1.16, 3.64) | |
| **Severity of anxiety symptoms at step 1 (GAD2 Score)** | 1.29 | (1.11, 1.49) | <0.01 |

**Table 4. Characteristics of older victims in the qualitative interviews.**

| P. No. | Crime Type | Sought Help From GP | Gender | Age | Ethnicity | Previous Anx/ Dep | Living Alone | Positive GAD/ PHQ Step 1 | Positive GAD/ PHQ Step 2 |
|---|---|---|---|---|---|---|---|---|---|
| 1 | Burglary | NO | F | 70 | White British | YES | NO | NO | NO |
| 2 | Harassment | NO | M | 71 | White British | NO | NO | YES | YES |
| 3 | Attempted burglary | NO | M | 74 | White British | NO | NO | YES | NO |
| 4 | Burglary | NO | M | 70 | White British | NO | NO | NO | NO |
| 5 | Theft from motor vehicle | YES | M | 71 | Asian Indian | YES | YES | YES | YES |
| 6 | Distraction Burglary | NO | F | 82 | White British | YES | NO | YES | YES |
| 7 | Distraction Burglary | YES | F | 76 | White British | YES | YES | YES | YES |
| 8 | Theft from person | YES | F | 65 | White British | YES | NO | YES | YES |
| 9 | Theft from motor vehicle | NO | M | 71 | White British | NO | NO | YES | NO |
| 10 | Theft from person | NO | F | 78 | Black African | NO | YES | YES | NO |
| 11 | Criminal damage to property—under £500 | NO | F | 67 | White British | YES | NO | YES | NO |
| 12 | Theft from motor vehicle | YES | F | 68 | Black African | NO | YES | NO | NO |
| 13 | Harassment | YES | M | 65 | Asian Indian | YES | YES | YES | YES |
| 14 | Fraud | NO | F | 83 | White British | NO | YES | YES | NO |
| 15 | Fraud | NO | M | 87 | White British | NO | YES | YES | NO |
| 16 | Theft from motor vehicle | NO | M | 76 | White–other | NO | NO | YES | NO |
| 17 | Common assault–racially/ religious aggravated assault | NO | M | 66 | White British | NO | YES | YES | NO |
| 18 | Actual bodily harm (ABH) | NO | F | 70 | White–other | NO | NO | YES | YES |
| 19 | Robbery of personal property with intimidation with knife | NO | M | 72 | Other ethnic group | NO | NO | NO | NO |
| 20 | Theft | NO | F | 83 | White British | NO | YES | NO | NO |
| 21 | Theft in a dwelling with intimidation | NO | F | 73 | White British | YES | YES | YES | YES |
| 22 | Burglary | YES | F | 94 | White British | | | NO | NO |
| 23 | Actual bodily harm (ABH) | YES | F | 65 | Black–Caribbean | YES | YES | YES | YES |
| 24 | Common assault–racially/ religious aggravated assault | NO | M | 68 | Mixed–other | NO | NO | YES | NO |
| 25 | Burglary | NO | F | 71 | Asian Indian | NO | YES | YES | YES |
| 26 | Fraud | NO | F | 91 | White British | YES | NO | YES | YES |
| 27 | Theft from motor vehicle | NO | F | 66 | White British | | YES | YES | YES |

Meanwhile, the participant who had been robbed at knifepoint felt that the police had given him more assistance than he needed: "*A man and a woman came to see how I was coping, then there was the follow-up call, I think they would have given me a psychologist if I'd said, 'I'm not doing very well', I said 'look I'm absolutely fine'*" (P19). This participant thought the police's perception of the severity of the crime may have influenced the amount of support he was offered: "*It may have been magic word 'knife,' of course, because that's the rage at the moment.*"

**Perception of services.** *Accessibility.* We asked participants what they thought would happen if they approached their GP. Many did not think that effective help would be provided. Some thought this was because doctors are not trained to deal with emotional problems: "*I think a lot of doctors would say 'this is not my job.' They might pass you on to a counsellor, but I think most GPs feel that they're not trained for such things.*" (P16). Others felt that doctors would not see it as a priority compared to physical health: "*I don't think I would expect much help because the doctors are under enormous pressure just to do the medicine. And I doubt that very many would have much ability to help with the problem for those people who are upset.*"

**Table 5. Thematic map of barriers and facilitators to seeking help.**

| | | | |
|---|---|---|---|
| Perception of themselves | | I am to blame for what happened | Barrier |
| | | I am too ashamed to tell people what happened | Barrier |
| | | You have to cope on your own | Barrier |
| | | I am too independent | Barrier |
| Perception of the crime | | Other crimes are more deserving of help than the crime I suffered | Barrier |
| | | Other victims have it worse than I do | Barrier |
| | | This should not be affecting me | Barrier |
| | | Services will not think the crime is severe enough | Barrier |
| Perception of services | Perception of service accessibility | Doctors are not trained for emotional problems | Barrier |
| | | Doctors will not see what happened as a priority | Barrier |
| | | I see my physical health as the priority | Barrier |
| | | Waiting lists are too long | Barrier |
| | | I do not know my GP well | Barrier |
| | | GPs are too busy to be sympathetic | Barrier |
| | | I am seeing my GP anyway | Facilitator |
| | | The crime is impacting on my physical health | Facilitator |
| | | I have a good relationship with my GP | Facilitator |
| | Perception of service acceptability | GPs only prescribe medication | Barrier |
| | | Support through social networks / faith communities is preferable | Barrier |
| | | GPs are people who do not have friends and family | Barrier |
| | | I have a good relationship with my GP | Facilitator |
| | | It is nice to have someone to talk to | Facilitator |
| | | I am spooked by the idea of therapy | Barrier |
| | | Talking with others who have had the same experience | Facilitator |
| | | Bring the support to older people | Facilitator |

(P15). Prioritising physical health was a particular concern for older victims, many of whom had complex health needs: "*I go often for my serious health complaints, I have a lot of other things to deal with, if my life were different perhaps* [I'd go to the GP about the crime] *but no, not at this time for me*" (P27). The need to prioritise physical health was attributed to demand on GP resources and difficulties obtaining appointments: "*If you're told when you phone the doctor that you're 20th in the queue and that the doctor can see you in 3 weeks' time or whatever.* [The letter] *is a lovely idea in principle. In practice, I don't honestly think that most doctors would see it as a priority*" (P1).

Some participants felt demands on GP time had changed their relationship with their doctor: "*You just don't get a chance to establish a rapport. Back in the 70s, it was the same doctor for about two decades, we were on first name terms*" (P17). As a result, they did not believe that GPs would be receptive to discussing the crime with them: "*GPs are so busy, I get the feeling you go in there like a conveyer belt, 'what's your problem?' 'This, this, this.' They hardly look at you, just look at the computer. 'Give you this, take that, next one please.' So, you don't get a lot of sympathy*" (P16). However, an exception to this was a participant (P13) who felt his health condition was being exacerbated by continued harassment from his neighbour. As he was visiting his GP regularly about his physical health, and saw the crime as connected, he was able to raise this during his appointments. He provided positive examples of the practical support his GP had offered: "*I did* [give the letter to the GP]. *My GP is very aware of all the problems I've had in this building because she's written letters of support to the local authority to move me out of here*" (P13). Help-seeking for psychological distress for this participant was therefore facilitated by frequent and positive contact with the same GP about the impact to his physical health.

*Acceptability.* Other participants were concerned about the type of help their GP may offer. Some expected they would be prescribed medication, which they felt was undesirable: *"I don't fancy running around on antidepressants, and thing is, if one goes to the doctor, invariably they will try to deal with the situation in that way."* (P21). Others felt that medication was for people who did not have social support: *"Medication is when the person has not got a big family"* (P2).

It was clear that many preferred to seek support informally instead of from professional services. For some, this was through friends or family: *"Having relatives that you know you can go to for help if you need it, you know. . .That means a lot, that you've got somebody there." (P22).* For others, it was faith communities: *"If we need support, we talk to certain circles of our friends within the church community"* (P4). Being able to talk was seen as beneficial: *"The more you talk about it, and get it out your system, it goes away quicker"* (P2). Of the participants who did seek professional support, having someone who was approachable was often the most helpful aspect: *"[The GP] was pretty sympathetic. He's the type of guy we can talk to"* (P6). Likewise, a participant who approached Victim Support commented: *"I could talk about it to get it off my system. . . cos certain people you can talk to about it, there are certain people you can't. . . but with your kind of people, you understand how I feel"* (P8). Yet despite many acknowledging the benefits of talking, some older people expressed doubts around accessing therapy: *"I'm spooked by the idea of what therapy is. . . I know it's being more public now talking about mental health"* (P9).

Finally, we asked participants what support would be helpful for other older crime victims. Again, having someone to talk to was a consistent theme. Victim Support was suggested: *"They're concerned, and they help you"* (P17). Others suggested support groups *"with others who have been through something similar"* (P21). The opportunity to discuss the experience with people who could personally relate to their experience appeared particularly helpful: *"One of the most important things is to meet other people who have had the same experience. . . I think it's helpful if you get people talking not with some expert, but with other people who've had a similar problem"* (P15). Others felt that bringing support to older victims was important: *"Go where older people go. I mean, the library, have a sign that you can come and talk"* (P17). Another suggested support could be brought to people in their homes: *"I think the best thing would be to go to someone's house. 'Cause that is where a person feels easiest"* (P6). This suggests bringing services to older victims may be more effective in engaging them in support, rather than older victims seeking out help themselves.

## 4. Discussion

Our study is the first to examine help-seeking for psychological distress in older crime victims. Despite our finding that over half of older victims suffer continued distress three months after a crime, there have been few attempts to understand service uptake or how services can be improved for this population.

Our first aim was to determine whether training the police to screen older victims for psychological distress within a month of a crime and provide letters signposting them to their GP was sufficient to improve primary care uptake. Our findings are consistent with The Filter Model [11], which describes how many people may suffer from a condition yet only few receive treatment. Our police screening was effective in detecting initial distress, but despite still suffering symptoms three months later, most (85%) did not visit their GP due to either not remembering the letter or deciding not to act on it. This is consistent with a growing number of studies which have found older adults [7] and crime victims of all ages [8] are reluctant to seek mental health support. Even when screen positive older victims did seek support in our study, only a third received treatment from their GP, despite our letters stating they were

suffering significant depressive and anxious symptoms. This suggests GPs need greater awareness and guidance on how to treat older crime victims, consistent with research on treating psychological symptoms attributed to social factors [22]. Taken together, signposting older victims to their GP is not sufficient to ensure they receive treatment.

Our second aim was to determine potential predictors of help-seeking behaviours in older victims. Caution is required when interpreting predictors of signposting as less than 13% of people acted on this. Furthermore, a significant number of predictors were included in the analysis, which may have increased the chance of a type 1 error. The aim was to explore potential predictors of help-seeking behaviours (association) instead of causes of help-seeking behaviours (causality) [23]. However, as the potential predictors were chosen using univariable logistic regression models and p-values, it is possible that variables indirectly associated with help-seeking behaviours may not have been selected.

Nevertheless, we found that participants who were identified by the police as being vulnerable and participants with more severe anxiety symptoms at step 1 were more likely to seek support from the GP. The police identify people as vulnerable on the Crime Reporting Information System for multiple reasons (physical, financial, emotional, neglect etc). Vulnerable older victims may be more likely to act on signposting because they already engaged with services or anxiety may have motivated them to seek help from their GP. By contrast, depression may be associated with hopelessness and a lack of motivation to seek help. We recommend training to improve identification and accuracy of recording vulnerability to ensure older victims in need are actively followed-up. Further research is required into the predictors of help seeking in older crime victims.

Our third aim was to qualitatively understand older victims' perceptions of the barriers and facilitators to help-seeking. When asked, most thought the GP letter could help others, but not themselves, which may be socially desirable responding [24] or because people are often better able to recognise the needs of others over themselves [25]. We identified several barriers which have previously been reported in other older adult studies. Many older victims felt they should 'get on with it', which is consistent with findings that older people view mental health issues as a normal part of ageing [26]. The need to maintain independence is consistent with trauma studies and ageing [12]. There appear to be contradictory findings with the suggestion that resilience is higher in older populations and age is a contributing factor to vulnerability [27]. Vulnerability has previously been considered a consequence of old age [28]. However, this may be more complicated, with recent research suggesting older adults do not consider themselves as vulnerable [29] despite being at greater risk of harm from victimisation [30]. Concerns around GP resources as a reason not to seek help has also been found in other research [31] and health campaigns urging the public to stay at home to protect healthcare services during Covid-19 [32] may still mean more older victims do not seek help for this reason.

Whilst many barriers can be applied to older adults more broadly, there were some which were specific to older crime victims. Self-blame and embarrassment are common in older victims [33], especially after fraud [34]. Feeling incompetent because of conditions associated with ageing, and attributing the crime to this, is an important barrier to be aware of in this group. Many did not think their crime was serious enough to justify help despite feeling distressed. As only a third of GPs treated distressed older victims who presented to them, doctors may consciously or unconsciously share this view. Basing treatment decisions on perceived crime severity may not be appropriate for older victims as concurrent life events, including declining physical health, depleting social networks and reduced income in retirement [35], can greatly increase the impact. For example, theft of a blue badge may be perceived as less severe given its low monetary value but in an older victim with mobility difficulties, it is likely to impact on their independence and quality of life [36].

Having a positive existing relationship with their GP was a facilitator to help-seeking in older crime victims, consistent with help-seeking predictors in other mental health populations [37]. Therapeutic relationships with GPs are likely to be impaired by recent changes in primary care structures. Practices are becoming increasingly large and individuals less likely to see the same doctor [38] and consultations are less likely to be face-to-face. Pressures on GP availability mean that where both physical and mental health problems were present, older people tended to prioritise physical health over mental health [39]. This was consistent with our finding that older victims appeared more receptive to help-seeking if they perceived the crime as impacting on their physical health. We identified a positive example of a GP writing a support letter to a housing association because the older victim felt the stress from the crime was aggravating existing disorders.

Many older victims thought talking about the experience would be helpful. This suggests talking therapies, such as cognitive behavioural therapy or acceptance and commitment therapy, might be acceptable treatments in older victims. This is consistent with other studies which have found older people prefer psychological therapies and do not like taking medication [40]. We also identified informal help-seeking in older victims from social networks or faith communities [9]. Many participants felt that bringing support to older victims–through the library, support groups, or home visits–would be more effective than primary care. Taking this in conjunction with our findings on the ineffectiveness of signposting letters and barriers to help-seeking, suggests that a model of assertive outreach is needed to help older victims after a crime.

## Strengths and limitations

This is the first study to collaborate with the police to assess signposting in older victims through quantitative screening using standardised measures and qualitatively exploring help-seeking behaviour in a subset.

A strength of our study is that police Safer Neighbourhood Team officers signposted over a thousand older victims. However, as 60% of crimes go unreported [41] our sample may not be fully representative of all older victims. Contacting the police may be a form of help-seeking in itself and the reasons for doing so may be varied, for example, to complete an insurance claim. Research also shows that older adults from ethnic minorities are less likely to report their crime to the police and report negative experiences of services [8]. It was also not possible to include older victims of domestic violence or sexual assault who may have different help-seeking behaviours. Nevertheless, given the challenges of timely identification of older victims, collaborating with organisations such as police is an important first step [1].

As we collected signposting data on 58% (677/1170) of initially screen positive participants, it is unknown whether people who could not be followed up acted on the signposting or not. However, our quantitative and qualitative samples have a similar demographic of older adults in North and East London [3] and findings appear to reflect our target population.

Quantitative and qualitative data suggested that 29.9% and 17% respectively of our population were from BME groups. Census data suggests that 16.9% of older people in London are of BME origin [4]. BME populations are often excluded from services. It is reassuring that our approach of targeting boroughs with high BME populations ensured good representation of these groups to enable us to identify the strengths and barriers to accessing care. We are also aware that with changing demographics, it is likely that non-white older victims will increase with time.

We used inductive qualitative analysis so were guided by the data and reported what the interviewees considered relevant. Some of the barriers and facilitators may be present in people of all ages, and also associated with cultural and physical factors specific to an older population.

The role of GPs is to help with mental health and physical problems. Because of the frailty associated with older age, older people are high utilisers of General Practice services and testing the role of signposting to GPs was considered the first step in deciding how to manage older victims. The recommendations made by older people to consider referral to voluntary and faith organisations may also be applicable to a younger population as part of victim care.

## Recommendations

We would recommend embedding support into local community services as signposting older victims to their GP is not sufficient. This would also be consistent with advice from the Health

**Table 6. Recommendations for managing older victims of crime.**

| | |
|---|---|
| Increasing knowledge of the impact of crime | Information campaigns outlining how psychological distress after a crime can impact on physical health may encourage older victims to come forward for help. This approach may help to overcome the physical and social barriers contributing to preventing older adults from accessing mental health support when needed [42].<br>This information should be distributed in settings which older people are likely to go to e.g. day centres, social clubs, and libraries. |
| Increasing referral/ self-referral | Encouraging victims to speak to a friend, family member, or faith communities initially and also involving these supports to encourage the victim to seek more formal help may be appropriate. Leaders of faith communities, day centres, social clubs could also be trained to identify the impact of crime and encourage seeking support. Having support groups set up in these venues may assist with identifying and making these referrals. |
| Case identification using structured screening tools | Whilst signposting letters were not largely effective, training the police to screen older victims for depression and anxiety as a means of early detection worked well. Instead of signposting to GPs, the police could be trained to support older victims to self-refer to Improving Access to Psychological Therapies (IAPT) services; a service which offers short-term psychological therapies to those suffering with anxiety, depression, or stress. |
| Increasing likelihood of action by health care professionals | GPs could enquire about crime experiences and impact as part of routine healthcare however, due to resource limitations, this may not always be feasible. If they are informed of the crime, GPs may be able to address specific concerns with the older victim. Promising findings have been reported for district nurses embedding care for older victims of distraction burglary, although further evaluation of this is needed [43].<br>The Health and Care Act is looking at addressing this by understanding how services can be integrated to provide better care. |
| | Of the older victims who do seek help after a crime, GPs should be mindful not to make unconscious judgements about the severity of the crime and treat older victims based on their clinical presentation. GPs may consider social prescribing, referrals to voluntary sector agencies such as Mind and Age Concern, or referral to IAPT. |
| | As older victims express a preference for talking treatments, we recommend psychotherapies such as cognitive behavioural therapy. The Victim Improvement Package, if found effective, could be rolled out by issuing government guidance suggesting a continuation of screening by SNTs. |
| | Demands on GP time and older victims' perceptions of accessibility and acceptability of help suggest that primary care is not the most effective means of supporting older victims. Assertive outreach has shown to be beneficial in supporting psychosocial needs of victims of crime which may be neglected through traditional support services and therefore, may be a valuable way of engaging older crime victims [7, 43]. |
| Further research | Whilst both the HAVoC study [7] and the current VIP trial have made progress towards implementing this by proactively offering CBT to victims of crime, further research is needed to understand how better to facilitate support for older victims. |

and Care Act [13] to integrate healthcare with public services. We also suggest raising GP awareness of the impact of crime on older victims and how they should best be managed, as outlined in Table 6.

## 5. Conclusion

Signposting does not appear to be effective in older crime victims and active follow-up of distressed older victims is recommended.

## Acknowledgments

We would like to thank Mr Dylan Peters and Ms Chloe Kirby for supporting the analysis. With special thanks to the Metropolitan Police staff: Chief Inspector Paul Ford, Inspector Martin Allen, Sergeants Gary Willis and Tim Pescud. We also wish to thank Paul Clarke, Head of the Organisation Learning and Research Faculty. We also extend our appreciation to Commanders: Bennet, Jerome, d'Orsi, Musker, Hidari and Watson, as well as members of the Metropolitan Police Safer Neighbourhoods Teams who made this research possible and the older victims of crime who gave up their time to take part in this study.

## Author Contributions

**Conceptualization:** Marc Serfaty, Jessica Satchell.

**Data curation:** Jessica Satchell.

**Formal analysis:** Victoria Vickerstaff, Teresa Lee, Jessica Satchell.

**Funding acquisition:** Marc Serfaty, Marta Buszewicz.

**Investigation:** Marc Serfaty.

**Methodology:** Marc Serfaty, Jessica Satchell.

**Project administration:** Jessica Satchell.

**Supervision:** Marc Serfaty, Jo Billings, Victoria Vickerstaff, Jessica Satchell.

**Writing – original draft:** Marc Serfaty, Jessica Satchell.

**Writing – review & editing:** Marc Serfaty, Jo Billings, Victoria Vickerstaff, Teresa Lee, Marta Buszewicz, Jessica Satchell.

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
