## [Decision Letter · Decision Letter 0]

2 Apr 2024

PMEN-D-23-00075

Help-Seeking in Older Crime Victims: A Mixed Methods Study in Collaboration with the Metropolitan Police Service

PLOS Mental Health

Dear Dr. Satchell,

Thank you for submitting your manuscript to PLOS Mental Health. After careful consideration, we feel that it has merit but does not fully meet PLOS Mental Health’s publication criteria as it currently stands. Therefore, we invite you to submit a revised version of the manuscript that addresses the points raised during the review process.

We look forward to receiving your revised manuscript.

Kind regards,

Ricardo Ventura Baúto, PhD

Academic Editor

PLOS Mental Health

Journal Requirements:

1. Please update your online Competing Interests statement. If you have no competing interests to declare, please state: “The authors have declared that no competing interests exist.”

2. In this instance it seems there may be acceptable restrictions in place that prevent the public sharing of your minimal data. However, in line with our goal of ensuring long-term data availability to all interested researchers, PLOS’ Data Policy states that authors cannot be the sole named individuals responsible for ensuring data access (http://journals.plos.org/mentalhealth/s/data-availability#loc-acceptable-data-sharing-methods).

3. Please provide separate figure files in .tif or .eps format only and ensure that all files are under our size limit of 10MB.

For more information about how to convert your figure files please see our guidelines: https://journals.plos.org/mentalhealth/s/figures

4. Tables should not be uploaded as individual files. Please remove these files and include the Tables in your manuscript file as editable, cell-based objects. For more information about how to format tables, see our guidelines: https://journals.plos.org/mentalhealth/s/tables

Additional Editor Comments (if provided):

Dear author,

Thank you for your submission.

We find the topic relevant and of great interest; however, some improvements are needed.

We recommend revising the English and clarifying some sentences, especially in the beginning of the manuscript, which can sometimes be difficult to read.

Other technical issues are listed below. We consider it important for you to review them, and then you can resend the manuscript, making appropriate references to the areas where changes were made.

Thank you.

Reviewer 1

This paper reports findings from a mixed methods study of the psychological impact of community crime on older victims. At least in gerontology, the focus is novel, although in the introduction the authors could make a stronger case for studying help seeking after crime in older adults. I have several questions that I hope the authors could address:

1. Page 5, Study Aims – Quantitative 1– Should it be “Help seeking” instead of “Health seeking behavior”?

2. Model building in the quantitative analysis appears to be completely data driven (ie based on p values). I am not convinced this is the best approach, even if the paper intends to be descriptive. With this approach, only the strongest associations would survive the selection, and they are usually not surprisingly tautologically. Some demographics, such as education, while may not relate to the outcome directly, often moderate the associations found, given known aged heterogeneity and its social patterning. Do the authors have any specific hypothesis as to how these variables may be related to help seeking?

3. The barriers and facilitators identified from the interviews are interesting, but not specific to older adults. This is fine, but one may wonder why the paper chose to focus on the older population and what the implications are for older adults.

Reviewer 2

It is a very interesting paper. However, it is not easy to read the first couple of pages of the paper due to the high number of acronyms but it is managable. Please explain acronyms like GP since not every non-native reader (as I am) knows that. I googled it but it would be better if you explain it in the first appearance.

Reviewers' comments:

Reviewer's Responses to Questions

**Comments to the Author**

1. Does this manuscript meet PLOS Mental Health’s publication criteria? Is the manuscript technically sound, and do the data support the conclusions? The manuscript must describe methodologically and ethically rigorous research with conclusions that are appropriately drawn based on the data presented.

Reviewer #1: Partly

Reviewer #2: Yes

2. Has the statistical analysis been performed appropriately and rigorously?

Reviewer #1: Yes

Reviewer #2: I don't know

3. Have the authors made all data underlying the findings in their manuscript fully available (please refer to the Data Availability Statement at the start of the manuscript PDF file)?

Reviewer #1: Yes

Reviewer #2: Yes

4. Is the manuscript presented in an intelligible fashion and written in standard English?

Reviewer #1: Yes

Reviewer #2: Yes

5. Review Comments to the Author

Reviewer #1: This paper reports findings from a mixed methods study of the psychological impact of community crime on older victims. At least in gerontology, the focus is novel, although in the introduction the authors could make a stronger case for studying help seeking after crime in older adults. I have several questions that I hope the authors could address:

1. Page 5, Study Aims – Quantitative 1– Should it be “Help seeking” instead of “Health seeking behavior”?

2. Model building in the quantitative analysis appears to be completely data driven (ie based on p values). I am not convinced this is the best approach, even if the paper intends to be descriptive. With this approach, only the strongest associations would survive the selection, and they are usually not surprisingly tautologically. Some demographics, such as education, while may not relate to the outcome directly, often moderate the associations found, given known aged heterogeneity and its social patterning. Do the authors have any specific hypothesis as to how these variables may be related to help seeking?

3. The barriers and facilitators identified from the interviews are interesting, but not specific to older adults. This is fine, but one may wonder why the paper chose to focus on the older population and what the implications are for older adults.

Reviewer #2: It is a very interesting paper. However, it is not easy to read the first couple of pages of the paper due to the high number of acronyms but it is managable. Please explain acronyms like GP since not every non-native reader (as I am) knows that. I googled it but it would be better if you explain it in the first appearance.

6. PLOS authors have the option to publish the peer review history of their article (what does this mean?). If published, this will include your full peer review and any attached files.

**Do you want your identity to be public for this peer review?** For information about this choice, including consent withdrawal, please see our Privacy Policy.

Reviewer #1: No

Reviewer #2: No

---

## [Editor Report · Decision Letter 1]

25 Jun 2024

Help-Seeking in Older Crime Victims: A Mixed-Methods Study in Collaboration with the Metropolitan Police Service

PMEN-D-23-00075R1

Dear Ms Satchell,

We are pleased to inform you that your manuscript 'Help-Seeking in Older Crime Victims: A Mixed-Methods Study in Collaboration with the Metropolitan Police Service' has been provisionally accepted for publication in PLOS Mental Health.

Best regards,

Ricardo Ventura Baúto, PhD

Academic Editor

PLOS Mental Health